# E-Cadherin Is Expressed in Epithelial Cells of the Choroid Plexus in Human and Mouse Brains

Genta Takebayashi [1,2], Yoichi Chiba [1], Keiji Wakamatsu [1], Ryuta Murakami [1], Yumi Miyai [1], Koichi Matsumoto [1], Naoya Uemura [2], Ken Yanase [2], Gotaro Shirakami [2], Yuichi Ogino [2] and Masaki Ueno [1,*]

[1] Department of Pathology and Host Defense, Faculty of Medicine, Kagawa University, 1750-1 Ikenobe, Miki-cho, Kita-gun, Takamatsu 761-0793, Kagawa, Japan; takebayashi.genta@kagawa-u.ac.jp (G.T.); chiba.yoichi@kagawa-u.ac.jp (Y.C.); s20d727@kagawa-u.ac.jp (K.W.); murakami.ryuta@kagawa-u.ac.jp (R.M.); miyai.yumi@kagawa-u.ac.jp (Y.M.); matsumoto.koichi@kagawa-u.ac.jp (K.M.)

[2] Department of Anesthesiology, Faculty of Medicine, Kagawa University, 1750-1 Ikenobe, Miki-cho, Kita-gun, Takamatsu 761-0793, Kagawa, Japan; uemura.naoya@kagawa-u.ac.jp (N.U.); yanase.ken@kagawa-u.ac.jp (K.Y.); shirakami.gotaro.bq@kagawa-u.ac.jp (G.S.); ogino.yuichi@kagawa-u.ac.jp (Y.O.)

* Correspondence: ueno.masaki@kagawa-u.ac.jp; Tel.: +81-87-891-2115

**Abstract:** Evidence showing the functional significance of the choroid plexus is accumulating. Epithelial cells with tight and adherens junctions of the choroid plexus play important roles in cerebrospinal fluid production and circadian rhythm formation. Although specific types of cadherin expressed in adherens junctions of choroid plexus epithelium (CPE) have been examined, they remained uncertain. Recent mass spectrometry and immunolocalization analysis revealed that non-epithelial cadherins, P- and N-cadherins, are expressed in the lateral membrane of CPE, whereas E-cadherin expression has not been confirmed in CPE of humans or mice. In this study, we examined E-cadherin expression in CPE of mice and humans by RT-PCR, immunohistochemical-, and Western blotting analyses. We confirmed, by using RT-PCR analysis, the mRNA expression of E-cadherin in the choroid plexus of mice. The immunohistochemical expression of E-cadherin was noted in the lateral membrane of CPE of mice and humans. We further confirmed, in Western blotting, the specific immunoreactivity for E-cadherin. Immunohistochemically, the expression of E- and N-cadherins or vimentin was unevenly distributed in some CPE, whereas that of E- and P-cadherins or β-catenin frequently co-existed in other CPE. These findings indicate that E-cadherin is expressed in the lateral membrane of CPE, possibly correlated with the expression of other cadherins and cytoplasmic proteins.

**Keywords:** adherens junction; choroid plexus epithelium; E-cadherin; N-cadherin; P-cadherin

## 1. Introduction

It is well-known that the choroid plexus epithelium (CPE) plays a central role in the secretion of cerebrospinal fluid by transport of ions and solutes from the blood supply to the brain parenchyma [1,2]. When the transport function of CPE was impaired, beta-amyloid clearance was reported to also be impaired in a triple transgenic mouse model of Alzheimer's disease [3]. Recently, it was found that the choroid plexus clock adjusts the suprachiasmatic nucleus clock, likely via circulation of cerebrospinal fluid, and regulates behavioral circadian rhythms [4]. The circadian rhythm is known to be disrupted in patients with senile dementia of the Alzheimer's type [5]. Several kinds of morphological or molecular abnormalities have been reported in the choroid plexus of human brains [6–8]. Calcified psammoma bodies are more likely to appear with aging in the choroid plexus stroma [6,7]. With liquid chromatography/mass spectrometry, Pearson et al. [8] identified alterations in the remodeling of epithelial adherens junction as well as mitochondrial bioenergetics and oxidative stress in tissues isolated from the walls of the inferior horn of the lateral ventricles of Alzheimer's disease compared with those of age-matched controls.

However, it remains to be clarified whether abnormalities in junctional proteins between epithelial cells in the choroid plexus contribute to decreased cerebrospinal fluid production and an impaired circadian rhythm.

Ions and water channels in the plasma membrane of CPE play a significant role in cerebrospinal fluid production [9,10]. Tight junctions have been investigated for many years and are now well-known to play a role in the transport permeation of monovalent cations as well as $H_2O$. The tight junctions contain both occludin and claudins and join associated cytosolic ZO-1 along the entire lateral surface [11]. Caudin-1, -2, -3, and -11 have all been demonstrated in the tight junctions of CPE [12–14]. Especially, claudin-2, a component of the tight junction, is known to form a paracellular water channel [15]. On the other hand, there is still controversy regarding adherens junctions. The expression of cadherins in the adherens junctions of CPE has been reported in many papers. However, specific forms of cadherin expressed in the central nervous system have not been fully determined and are still debatable [9,16–19]. Lagunowich et al. [16] reported that N-cadherin is expressed at high levels in a uniform fashion in many regions of the central nervous system during histogenesis and, in addition, that N-cadherin expression becomes restricted to ependymal cells and CPE during later stages of development. Kaji et al. [18] reported that P-cadherin is expressed on the basal side of CPE in mice. Christensen et al. [9], using mass spectrometry analysis and RT-PCR, reported that P- and N-cadherins were both localized to the lateral membrane of the CPE of mice.

Concerning the expression of E-cadherin, Figarella-Branger et al. [20] reported that anti-E-cadherin immunoreactivity (British Biotechnology Products, clone HEC1) was observed on the basolateral part of most adult CPE in the third and fourth ventricles in surgically removed tissues diagnosed as benign ependymoma and benign choroid plexus papilloma. However, in the manuscript, they stated the possibility that the monoclonal anti-E-cadherin antibody used in the study cross-reacts with other members of the cadherin superfamily. They stated that the cytoplasmic domains of B-cadherin and E-cadherin were 88% identical in chickens, although B-cadherin expression was not reported in human CPE [20]. On the other hand, RT-PCR, Western blot analysis, and immunocytochemistry revealed that mRNA and proteins of E-cadherin were expressed in two immortalized epithelial cell lines derived from rat choroid plexus and human choroid plexus carcinoma cells, CPC-2 [21]. However, it was found that E-cadherin immunoreactivity was not localized to areas of cell-cell contact in CPC-2 cells, whereas it was localized to areas of cell-cell contact in rat cell lines [21]. Christensen et al. reported [19] that immunoreactivity of mouse monoclonal anti-E-cadherin antibody (610181, BD Biosciences, Franklin Lakes, NJ, USA) was localized to the basolateral membrane of the mouse CPE. However, Christensen et al., in a more recent paper [9], reported that the expression of E-cadherin in CPEs was neither detected by mass spectrometry analysis nor RT-PCR, consistent with some previous reports [16–18]. In a recent manuscript [9], they reported that the antibody for E-cadherin previously applied to localize to the mouse CPE [19] cross-reacts with P-cadherin, according to the manufacturer.

Thus, it remains unclear whether E-cadherin expression exists in the CPE of mouse brains or human brains without brain tumors. Accordingly, in this study, initially, we examined whether E-cadherin is expressed in the mouse CPE by RT-PCR, immunohistochemical, and Western blotting analyses. Then, we examined whether E-cadherin is immunohistochemically expressed in epithelial cells using choroid plexus located in the lateral ventricles of human brains without tumors, which has not yet been investigated, and whether E-cadherin is colocalized with N- or P-cadherin.

## 2. Materials and Methods

### 2.1. Animals

All animal studies were approved by the Kagawa University Animal Care and Use Committee (#21636), and all efforts were made to minimize the number and extent of suffering of animals used. Under deep anesthesia with the intraperitoneal injection of pentobarbital, 8 to 10-week-old male C3H/HeSlc mice (*n* = 11) (Japan SLC, Hamamatsu,

Japan) were transcardially perfused with phosphate-buffered saline (PBS), followed by the procedures described below for immunohistochemical, Western blotting, and RT-PCR analyses.

## 2.2. Human Tissues

Human tissue samples were obtained at autopsy in Kagawa University Hospital, as previously described [22,23]. Table 1 summarizes the clinical profiles of all subjects. This study conformed to the Declaration of Helsinki and was approved by the Institutional Ethics Committee of the Faculty of Medicine, Kagawa University (#H24-48).

**Table 1.** Summary of clinicopathological profiles.

| (No.) | Age/Sex | Main Diagnosis |
|---|---|---|
| 1 | 42/M | Pulmonary hypertension, Heart failure |
| 2 | 57/F | Psychiatric disorder, Liver abscess, Sepsis |
| 3 | 64/M | Multiple system atrophy, Pneumonia |
| 4 | 68/F | Thalamic hemorrhage |
| 5 | 70/M | Myocardial infarction |
| 6 | 72/F | Pneumonia |
| 7 | 74/M | Lung cancer |
| 8 | 75/M | Gastric cancer |
| 9 | 75/M | Multiple system atrophy, Pneumonia |
| 10 | 84/M | Myocardial infarction, Cerebral infarction |

## 2.3. Immunohistochemical Procedure in Mouse Brains

For immunohistochemical analysis, mice ($n = 5$) were perfused with PBS, followed by perfusion with 4% paraformaldehyde in 0.1 M phosphate buffer (pH 7.4). Dissected tissues were postfixed with the same fixative at 4 °C overnight, embedded in paraffin, and cut into 4-μm-thick sections. Paraffin sections from brain samples including choroid plexus of mice were deparaffinized and pretreated with 0.3% hydrogen peroxide in PBS for 30 min to block endogenous peroxidase activity. After blocking with 2% bovine serum albumin (BSA) in PBS for 30 min, the sections were incubated with antibodies at 4 °C overnight [22,23]. Primary antibodies against E-cadherin (mouse monoclonal, clone NCH-38; Dako, Glostrup, Denmark; 1:250) [24,25], E-cadherin (rabbit polyclonal, Cat. No. 20874-1-AP; ProteinTech Group, Chicago, IL, USA; 1:2000) [26,27], N-cadherin (rabbit polyclonal, Cat. No. 22018-1-AP; ProteinTech Group; 1:1000) [27,28], and P-cadherin (rat monoclonal, 13-2000Z, clone PCD-1; Invitrogen, ThermoFisher Scientific, Rockford, IL, USA) [29] were used. Information on the primary antibodies for mice is summarized in Table 2. Before incubation with some antibodies, antigen retrieval was performed by heating sections in 10 mM sodium citrate buffer (pH 6) or 1 mM Tris-ethylenediaminetetraacetic acid (EDTA) (pH 9) at 95 °C for 20 min, as shown in Table 2. After treatment with hydrogen peroxide and blocking with 2% bovine serum albumin in PBS for 30 min, the sections were incubated with primary antibodies at 4 °C overnight. Sections were washed with PBS and then incubated with a polymer solution conjugated with anti-rabbit IgG antibody, anti-mouse IgG antibody, or anti-rat IgG antibody and horseradish peroxidase (HRP) (Histofine® Simple Stain™ MAX PO, Nichirei Biosciences Inc., Tokyo, Japan) and developed with 3,3′-diaminobenzidine. The sections were counterstained with hematoxylin.

**Table 2.** Summary of antibodies used.

| Antibody | Cat. No. (Clone Name) | Host Species and Usage | References |
|---|---|---|---|
| E-cadherin | Dako, M3612 (NCH-38) | mouse, 1:250 (¶1) | [24,25] |
| E-cadherin | ProteinTech, 20874-1-AP | rabbit, 1:2000 (¶2) | [26,27] |
| N-cadherin | ProteinTech, 22018-1-AP | rabbit, 1:1000 (¶2) | [27,28] |
| P-cadherin | Invitrogen, 13-2000Z (PCD-1) | rat, 1:200 (¶2) | [29] |
| P-cadherin | Santa Cruz, sc-74545 (A-10) | mouse, 1:100 (¶1) | [28] |
| β-catenin | Santa Cruz, sc-7199 | rabbit, 1:50 (¶2) | [30] |
| vimentin | Dako, M0725 (V9) | mouse, 1:50 (¶1) | [6,23] |

(¶1, ¶2): Antigen retrieval with citrate buffer (pH 6) or Tris-EDTA buffer (pH 9) is needed prior to the application of the primary antibody, respectively. Dako Agilent (Santa Clara, CA, USA).

*2.4. Western Blotting in Choroid Plexus Tissues of Mice*

After perfusion with PBS, choroid plexus tissues isolated from the lateral and fourth ventricles of 8-week-old C3H/He male mice ($n$ = 3) were placed in cold PBS. The tissues were homogenized in 20 µL of radioimmunoprecipitation assay (RIPA) buffer containing protease inhibitor cocktail (PIC: Halt Protease Inhibitor Cocktail, Thermo Fisher Scientific, Waltham, MA, USA) using homogenization pestle for microcentrifuge tube (Scientific Specialities, Inc., Lodi, CA, USA). After centrifuging at 21,500× $g$ at 4 °C for 20 min, the supernatants were collected and the protein concentration was determined using a Pierce BCA Protein Assay Kit (Thermo Fisher Scientific).

Twenty µg of protein/lane was subjected to SDS-polyacrylamide gel electrophoresis (PAGE) on a 7.5% polyacrylamide gel along with a molecular mass marker solution (Ex-elBand 3-color Pre-Stained Protein Ladder, Broad Range: SMOBio, Hsinchu, China). The separated proteins were transferred to polyvinylidene difluoride membranes (FUJIFILM Wako Pure Chemical, Osaka, Japan) using semi-dry techniques. The membranes were stained with Commassie Brilliant Blue (CBB) (Rapid Stain CBB Kit, Nacalai Tesque, Kyoto, Japan) and scanned to quantify the amount of protein on blots. After destaining with a Rapid CBB Destain Kit (Nacarai Tesque, Kyoto, Japan), membranes were treated with 5% skim milk in Tris-buffered saline containing 0.1% Tween 20 (TBS-T) to block nonspecific antibody binding, and then probed with anti-E-cadherin antibody (Dako; 1:1000) at 4 °C overnight. After washing with TBST, blots were incubated with HRP-conjugated anti-rabbit IgG (ProteinTech, Rosemont, IL, USA; 1:1000). Primary and secondary antibodies were diluted in 'Can Get Signal Immunoreaction Enhancer Solution' (Toyobo, Osaka, Japan). The immunoreactive bands were visualized using ImmunoStar and a chemiluminescence imager (Amersham™ ImageQuant™ 800: Cytiva, Tokyo, Japan).

*2.5. RT-PCR Analysis of Choroid Plexus Tissues in Mice*

After perfusion with PBS, choroid plexus tissues in lateral and fourth ventricles, and small intestine tissues were isolated from mice ($n$ = 3) and total RNA was extracted using ReliaPrep™ RNA Tissue Miniprep System (Promega, Fitchburg, WI, USA). The cDNA was synthesized using ReverTra Ace® qPCR RT Master Mix (Toyobo, Osaka, Japan). Ten nanograms of cDNA were used as a template and amplified with PrimeSTAR® Max DNA Polymerase (TAKARA BIO Inc., Kusatsu, Japan) and specific primers: 3 primer sets for mouse Cdh1 were used in this study [22,31] (Table 3). PCR amplification cycle conditions were 24 (Gapdh) and 33 (Cdh1) cycles of 10 s at 98 °C, 5 s at 55 °C, and 5 s at 72 °C. The amplicons were electrophoresed on a 2% agarose gel, stained with Midori Green Advance (Nippon Genetics, Tokyo, Japan), and visualized with a blue LED (470 nm) transilluminator (AMZ System Science, Osaka, Japan). Amplified fragments were purified using Nucleospin® Gel and PCR Clean-up (MACHEREY-NAGEL GmbH & Co. KG, Düren, Germany) and subjected to direct sequencing (Eurofins Genomics, Tokyo, Japan).

**Table 3.** Primer sequences used for RT-PCR.

| Genes | Primers (5′-3′) | Location | Ampicon Size (bp) | Reference |
|---|---|---|---|---|
| *Cdh1* (mCdf1-1) | F: GGAGACCAGTTTCCTCGTCC<br>R: CATTTTCGGGGCAGCTGATG | 433–452<br>650–631 | 218 | |
| *Cdh1* (mCdf1-2) | F: GCTCTCATCATCGCCACAGA<br>R: GCAGTAAAGGGGGACGTGTT | 1838–1857<br>2034–2015 | 197 | |
| *Cdh1* (mCdh1-3) | F: CTGCCATCCTCGGAATCCTT<br>R: TGGCTCAAATCAAAGTCCTGGT | 2274–2293<br>2463–2442 | 190 | [31] |
| *Gapdh* | F: CAAGGTCATCCATGACAACTTTG<br>R: GTCCACCACCCTGTTGCTGTAG | 527–549<br>1022–1001 | 496 | [22] |

*2.6. Immunohistochemical Procedure in Human Brains*

Human brains were removed at autopsy and fixed with 10% neutral buffered formalin. Fixed human medial temporal lobes including choroid plexus in lateral ventricles were removed and embedded in paraffin. Paraffin-embedded tissue blocks were prepared and cut into 4-µm-thick sections. Initially, the sections were stained with Hematoxylin-Eosin (H&E). For immunohistochemical analysis of the human choroid plexus, paraffin sections were deparaffinized and pretreated with 0.3% hydrogen peroxide in PBS for 30 min to block endogenous peroxidase activity. After blocking with 2% bovine serum albumin (BSA) in PBS for 30 min, the sections were incubated with antibodies at 4 °C overnight [22,23]. Primary antibodies against E-cadherin (mouse monoclonal, clone NCH-38; Dako, Glostrup, Denmark; 1:250) [24,25], E-cadherin (rabbit polyclonal, Cat. No. 20874-1-AP; ProteinTech Group, Chicago, IL, USA; 1:2000) [26,27], N-cadherin (rabbit polyclonal, Cat. No. 22018-1-AP; ProteinTech Group; 1:1000) [27,28], P-cadherin (mouse monoclonal, sc-74545, clone A-10; Santa Cruz Biotechnology, Dallas, TX, USA; 1:100) [28], β-catenin (rabbit polyclonal, sc-7199; Santa Cruz Biotechnology; 1:50) [30], and vimentin (mouse monoclonal, clone V9; Dako, Glostrup, Denmark; 1:50) [6,23] were used. Information on the primary antibodies for human brains is summarized in Table 2. Before incubation with some antibodies, antigen retrieval was performed by heating sections in 10 mM sodium citrate buffer (pH 6) or 1 mM EDTA (pH 9) at 95 °C for 20 min, as shown in Table 2. After treatment with hydrogen peroxide and blocking with 2% bovine serum albumin in PBS for 30 min, the sections were incubated with primary antibodies at 4 °C overnight. Sections were washed with PBS and then incubated with a polymer solution conjugated with anti-rabbit IgG antibody or anti-mouse IgG antibody and horseradish peroxidase (HRP) (Histofine® Simple Stain™ MAX PO, Nichirei Biosciences Inc., Tokyo, Japan) and developed with 3,3′-diaminobenzidine. The sections were counterstained with hematoxylin.

For double immunofluorescence examination, the sections were incubated at 4 °C overnight with the mouse antibody for E-cadherin (Dako, clone NCH-38), P-cadherin (sc-74545), or vimentin (M0725) and the rabbit antibody for N-cadherin (22018-1-AP), E-cadherin (20874-1-AP), or β-catenin (sc-7199), followed by incubation at RT for 60 min with Alexa Fluor 488-anti-mouse IgG and Alexa Fluor 594-conjugated anti-rabbit IgG antibodies (Molecular Probes, Eugene, OR, USA; 1:200), respectively. Before incubation with primary antibodies, antigen retrieval was performed by heating sections in 1 mM Tris-EDTA (pH 9) at 95 °C for 20 min. The sections were then incubated for 60 min at RT in Monomeric Cyanine Nucleic Acid Stain (TO-PRO-3, Molecular Probes, Eugene, OR, USA), which was diluted to 2.5 µM in PBS. The fluorescent signals were viewed under a confocal microscope (Carl Zeiss LSM700, Oberkochen, Germany). As a control experiment, we performed an identical immunohistochemical procedure with the omission of the primary antibodies.

For morphometrical analyses in paraffin sections including choroid plexus of ten autopsied human brains, the percentage of epithelial cells immunostained with antibodies for E-cadherin (Dako), E-cadherin (Proteintech, Chicago, IL, USA), N-cadherin, P-cadherin (SantaCruz, Dallas, TX, USA), β-catenin, and vimentin was examined in ten randomly

selected areas and mean values were calculated in six groups of ten human brains. The correlation of mean values in ten human brains between two groups was analyzed by Peason's correlation coefficient test. $p < 0.05$ was considered statistically significant.

## 3. Results

### 3.1. Cadherin Expression in Mouse Choroid Plexus

Immunoreactivity for E-cadherin (Dako, M3612) was seen in the lateral membrane of CPE of all mice examined (Figure 1A). Western blotting analysis of the membrane fraction of 3 choroid plexus samples of mice with the anti-E-cadherin antibody (Dako, M3612) revealed the presence of a single immunoreactive band for E-cadherin with a molecular mass of approximately 120 kDa (Figure 1B). In addition, immunoreactivities for E-cadherin (ProteinTech), N-cadherin, and P-cadherin (Invitrogen, ThermoFisher Scientific, Waltham, MA, USA) were also seen in the lateral membrane of CPE of mice (Supplemental Figure S1). Immunoreactivities for E-, N-, and P-cadherins seemed to be evenly distributed on the lateral membrane in the CPEs of mice. RT-PCR analysis revealed that Cdh1 mRNA is expressed in choroid plexus tissues derived from the lateral and fourth ventricles of 3 mice (Figure 1C). There were no clear differences in mRNA expression levels of E-cadherin in choroid plexus samples between the lateral and fourth ventricles. However, the expression of Cdh1 mRNA in choroid plexus tissues was lower than that in the small intestinal tissue (Figure 1C). All 3 primer sets for Cdh1 (mCdh1-1, mCdh1-2, and mCdh1-3) amplified bands with modest intensities from cDNAs derived from the mouse choroid plexus (Supplemental Figure S2) and yielded similar results in 3 mice. Sequence analysis of the amplicons confirmed them to be the expected fragments of mouse Cdh1 (Supplemental Figure S2).

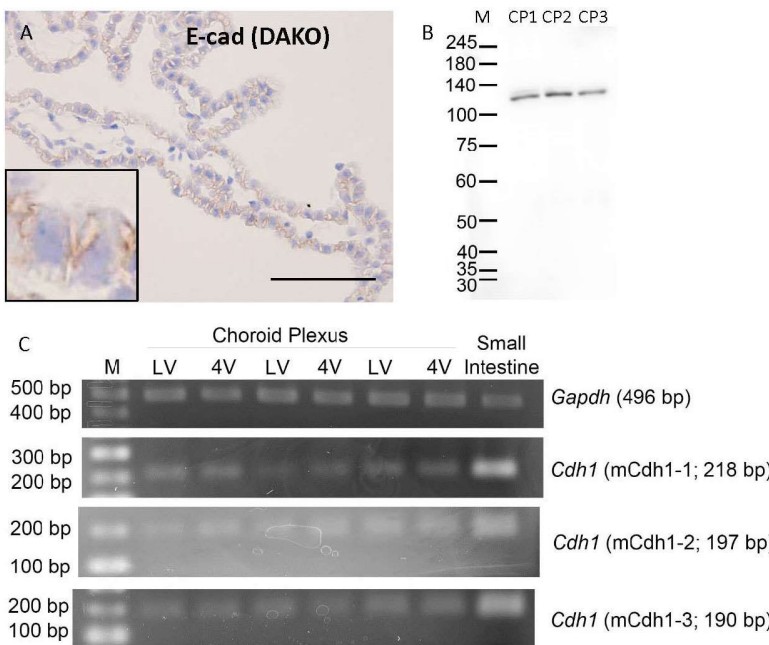

**Figure 1.** A representative microphotograph of immunohistochemical staining in choroid plexus (**A**), immunoblots of 3 choroid plexus samples (**B**) using the antibody against E-cadherin (Dako, M3612, NCH-38), and expression of Gapdh and Cdh1 mRNAs in choroid plexus and small intestine samples (**C**). (**A**); An inset in (**A**) shows an enlarged image of the microphotograph. Scale bars indicate 100 μm. (**B**); M: molecular marker, CP1-3: choroid plexus samples from 3 mice. (**C**); Expression of Cdh1 mRNA in mouse choroid plexus located in the lateral and fourth ventricles is shown using 3 sets of primers. Results for the choroid plexus from 3 mice and those for the small intestine from 1 mouse are presented. Amplicons obtained from RT-PCR for Gapdh are presented to show that an approximately equal amount of cDNA was added to each reaction tube. LV: lateral ventricle, 4V: fourth ventricle, M: DNA molecular weight marker. Three kinds of primers for Cdh1 mRNA amplification are used and shown in Table 3.

### 3.2. Cadherin Expression in Human Choroid Plexus

Immunoreactivities for E-cadherin using two-kinds of antibodies were seen in the lateral membrane of human CPE (Figure 2A,B). There was no difference in immunoreactivities between mouse monoclonal (Dako, M3612) and rabbit polyclonal (ProteinTech, 20874-1-AP) antibodies for E-cadherin (Figure 2A,B). Immunoreactivities for N- and P-cadherins were also noted in the lateral membrane of human CPE (Figure 2C,D). Immunoreactivity for β-catenin was seen in the lateral membrane of human CPE (Figure 2E), whereas that for vimentin was in the basolateral membrane of human CPE (Figure 2F). All H&E staining and immunohistochemical images for E-, N-, and P-cadherins in ten human brains are shown in Supplemental Figure S3. As reported in a previous paper [23], dense fibrous or calcified materials were seen in the stroma just below CPE in all autopsied human brains. Epithelial cells just above the calcified materials were frequently thin or absent [23]. Immunoreactivities for cadherins in those epithelial cells were sometimes weak or had disappeared.

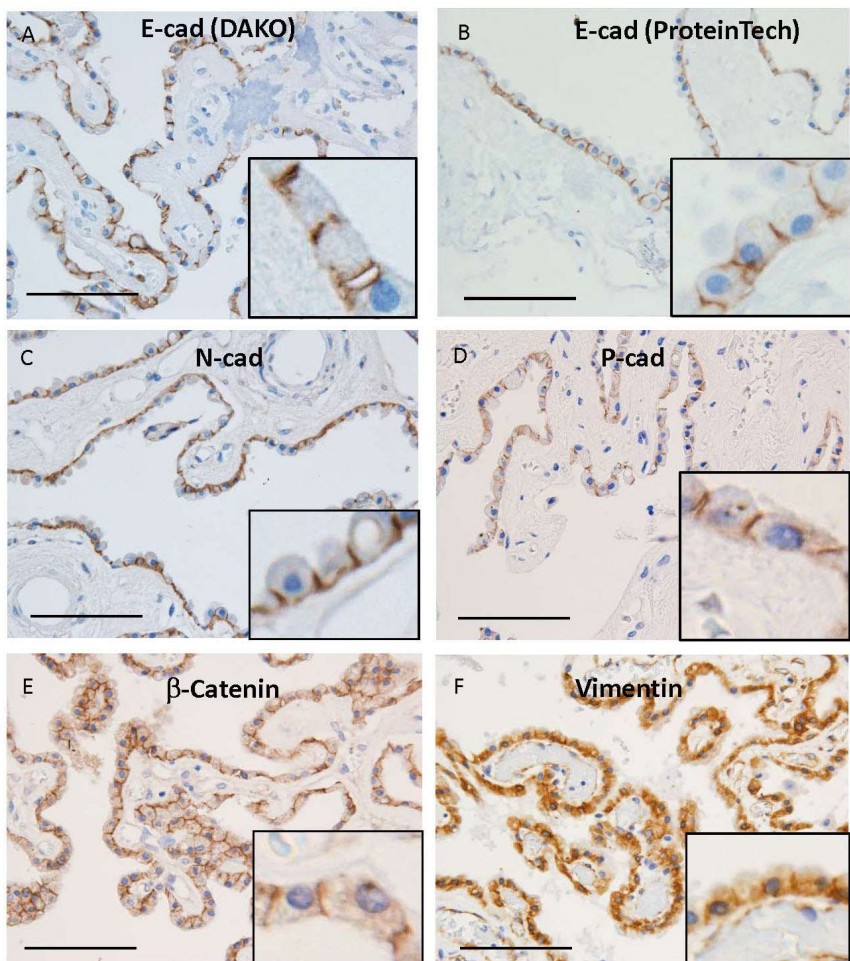

**Figure 2.** Representative microphotographs of immunohistochemical staining using antibodies against E-cadherin (E-cad) (DAKO, M3612, NCH-38) (**A**), E-cadherin (ProteinTech, 20874-1-AP) (**B**), N-cadherin (N-cad) (**C**), P-cadherin (P-cad) (Santa Cruz, sc-74545) (**D**), β-catenin (**E**), and vimentin (**F**). Insets of (**A–F**) show enlarged images of microphotographs. Images from cases 1 (**F**), 3 (**C**), 5 (**E**), 6 (**A**), 7 (**D**), and 8 (**B**) are shown. Scale bars indicate 100 μm.

### 3.3. Double Immunofluorescence Examination of Cadherin Expression

Double immunofluorescence examination using human brain samples showed that immunoreactivity for E-cadherin was colocalized with that for N-cadherin on the lateral membrane in some CPEs (Figure 3A–C). However, immunoreactivities for E- and N-cadherins were partially unevenly distributed on the lateral membrane of other CPEs

(Figure 3D–F). Immunoreactivities for E- or N-cadherin, indicated by dotted arrows in Figure 3D or Figure 3E, respectively, were not seen in the same plasma membranes in Figure 3E or Figure 3D. Immunoreactivities for E- and P-cadherins were mostly colocalized on CPE, whereas immunoreactivities of these cadherins were unevenly distributed on the lateral membrane in a few cells (Figure 3G–I). Immunoreactivity for P-cadherin was partially colocalized with that for N-cadherin on the lateral membrane in some CPEs (Figure 3J–L), whereas immunoreactivities for P- and N-cadherins were frequently unevenly distributed on the lateral membrane in other CPEs (Figure 3M–O). Immunoreactivities for P- or N-cadherin indicated by dotted arrows in Figure 3M or Figure 3N, respectively, were not seen in the same plasma membranes in Figure 3N or Figure 3M, respectively.

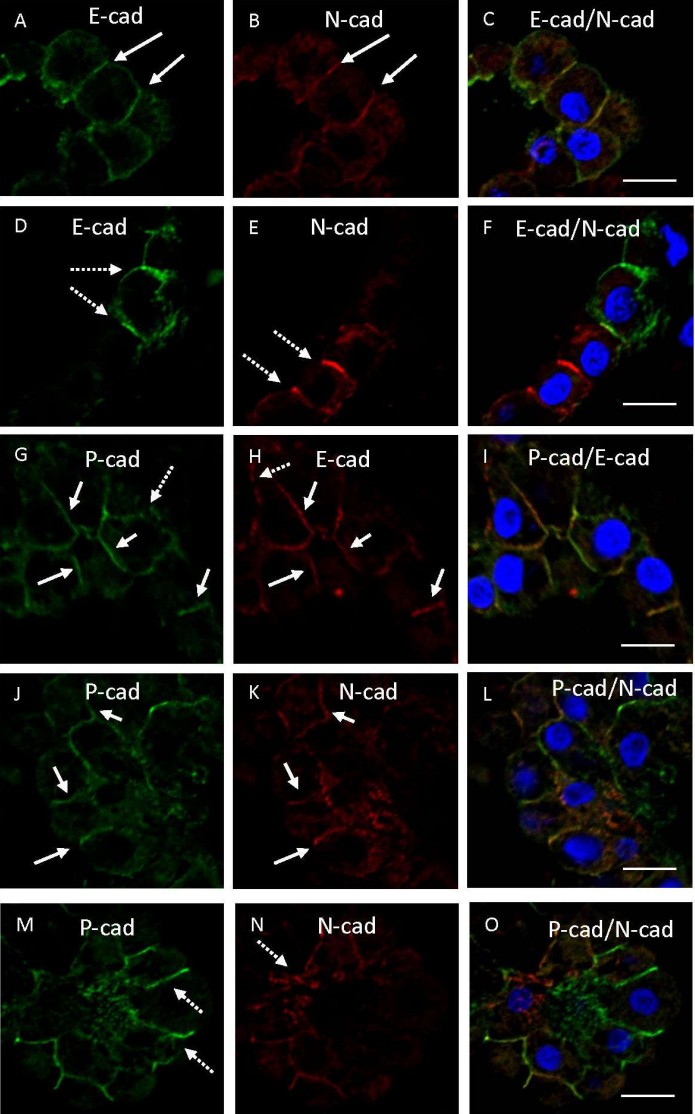

**Figure 3.** Representative confocal microscopic images showing double immunostaining in CPE. Immunostaining with the mouse anti-E-cadherin (Dako) (**A,D**) or P-cadherin (**G,J,M**) antibody (visualized as green) and the rabbit anti-N-cadherin (**B,E,K,N**) or E-cadherin (ProteinTech) (**H**) antibody (visualized as red), nuclear staining by TO-PRO-3 (visualized as blue), and merged images (**C,F,I,L,O**) are shown. Solid arrows in images (**A,B,G,H,J,K**) show double immunopositive substances, whereas dotted arrows show an uneven distribution of expressed substances (**D,E,G,H,M,N**). Images from cases 1 (**G–O**) and 10 (**A–F**) are shown. Scale bars indicate 10 μm.

Regarding the relationship of the immunohistochemical expression between E-cadherin and cytoplasmic markers, immunoreactivity for E-cadherin was mostly colocalized with

that for β-catenin on the lateral membrane in CPE (Figure 4A–C). On the other hand, immunoreactivities for E-cadherin and vimentin were sometimes unevenly distributed on the lateral membrane in some CPEs (Figure 4D–F). Immunoreactivities for vimentin or E-cadherin, indicated by dotted arrows in Figure 4D or Figure 4E, were not seen in the same plasma membranes in Figure 4E or Figure 4D, respectively.

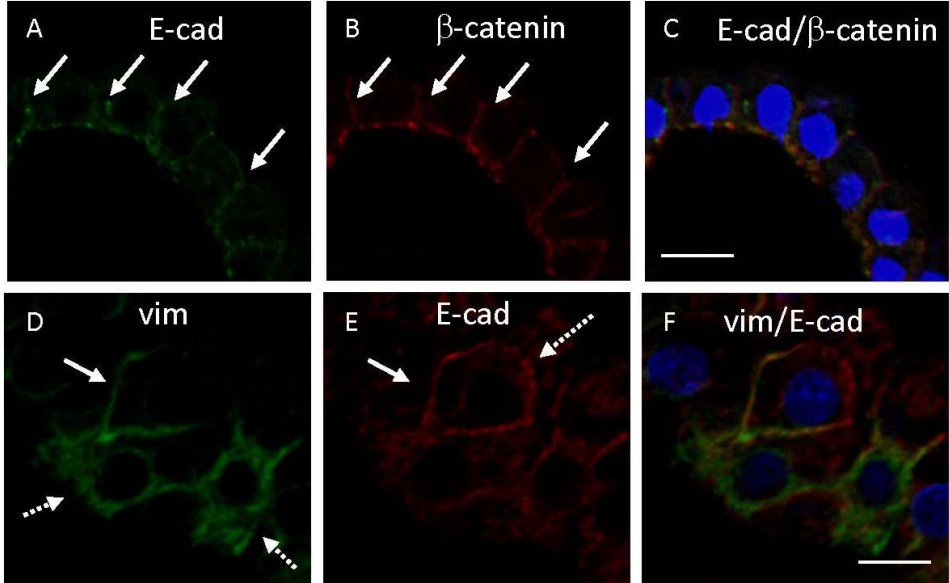

**Figure 4.** Representative confocal microscopic images showing double immunostaining in CPE. Immunostaining with the mouse anti-E-cadherin (Dako) (**A**) or vimentin (**D**) antibody (visualized as green) and the rabbit anti-β-catenin (**B**) or E-cadherin (ProteinTech) (**E**) antibody (visualized as red), nuclear staining by TO-PRO-3 (visualized as blue), and merged images (**C**,**F**) are shown. Solid arrows in images (**A**,**B**,**D**,**E**) show double immunopositive substances, whereas dotted arrows show an uneven distribution of expressed substances (**D**,**E**). Images from case 1 (**A**–**F**) are shown. Scale bars indicate 10 μm.

Mean values of the percentage of CPEs immunostained with antibodies for E-cadherin (Dako), E-cadherin (ProteinTech), N-cadherin, P-cadherin (SantaCruz), β-catenin, and vimentin in ten human brains are shown in Table 4. Peason's correlation coefficient tests showed a significant correlation between the ratios of immunostained cells for E-cadherin (Dako) and E-cadherin (ProteinTech) ($p < 0.01$), β-catenin ($p < 0.01$), or vimentin ($p < 0.05$) and between those for E-cadherin (ProteinTech) and β-catenin ($p < 0.05$). However, there was no correlation between the ratios of immunostained cells for E-cadherin and N- or P-cadherin using either antibody for E-cadherin.

**Table 4.** The ration of immunostained cells to total cells (percentage).

| (No.) | Age/Sex | A | B | C | D | E | F |
|---|---|---|---|---|---|---|---|
| 1 | 42/M | 79 | 91 | 81 | 31 | 98 | 94 |
| 2 | 57/F | 47 | 74 | 84 | 25 | 73 | 54 |
| 3 | 64/M | 68 | 81 | 57 | 29 | 77 | 98 |
| 4 | 68/F | 12 | 68 | 87 | 24 | 28 | 42 |
| 5 | 70/M | 73 | 94 | 84 | 38 | 96 | 98 |
| 6 | 72/F | 51 | 86 | 73 | 15 | 82 | 88 |
| 7 | 74/M | 64 | 95 | 46 | 19 | 86 | 70 |
| 8 | 75/M | 56 | 92 | 61 | 22 | 74 | 98 |
| 9 | 75/M | 43 | 68 | 67 | 40 | 72 | 99 |
| 10 | 84/M | 67 | 98 | 81 | 37 | 76 | 97 |

A–F indicate antibodies used for morphometrical analysis. A: E-cad (Dako), B: E-cad (ProteinTech), C: N-cad, D: P-cad (SantaCruz), E: β-catenin, F: vimentin.

## 4. Discussion

Immunoreactivity for E-cadherin in CPE in the lateral ventricle of autopsied human brains without epithelial or ependymal cell tumors has not been determined. Mass spectrometry, RT-PCR, and immunolocalization analyses confirmed that P- and N-cadherins were expressed in the mouse CPE, whereas it remains unclear whether E-cadherin is expressed in the mouse and human CPE. In this study, localization and expression of E-cadherin were examined in CPE of mouse and autopsied human brains using immunohistochemical, Western blotting, and RT-PCR methodologies. RT-PCR analysis using 3 sets of primers for *Cdh1* mRNA amplification revealed that mRNA of E-cadherin was confirmed to be expressed in small quantities in the mouse choroid plexus. Immunohistochemical and Western blotting examination using the monoclonal antibody for E-cadherin, which had already been used for immunohistochemistry and Western blotting analysis [24,25], was performed with choroid plexus tissues of mice. The results showed that immunoreactivity for E-cadherin was localized to lateral membranes of the CPE of mice. Western blotting confirmed the specificity of immunoreactivity for E-cadherin (Dako). In addition, immunoreactivities for E-cadherin were also seen in the lateral membrane of CPE in humans. These findings indicate that E-cadherin is expressed in lateral membranes of mouse and human CPE. On the other hand, Figarella-Branger et al. [20] stated the possibility that the monoclonal anti-E-cadherin antibody (British Biotechnology Products, clone HEC1) cross-reacts with other members of the cadherin superfamily. Immunopositive areas of P-cadherin were very similar to those of E-cadherin in this study. According to the alignment of amino acid sequences of E-cadherin and P-cadherin from humans and mice, their identities were 54% in humans and 52% in mice. The alignment of partial amino acid sequences of E-cadherin used as an immunogen was 54% identical to that of P-cadherin. Two kinds of antibodies for E-cadherin (Dako, M3612; ProteinTech, 20874-1-AP) have been widely used for immunohistochemistry as well as Western blot [24–27]. A single band around 120 kDa was detected by Western blot using E-cadherin antibody (M3612). In addition, RT-PCR analysis revealed that mRNA of E-cadherin is expressed in choroid plexus tissues of mice. Accordingly, the findings in this study indicate that E-cadherin is expressed in the lateral membrane of CPE of mice and humans. However, the possible cross-reactivity of cadherin antibodies cannot be completely excluded judging from the findings in this study alone.

There were no clear differences in the localization of immunoreactivities for E-, N-, and P-cadherins among the 10 autopsied human brains (Supplemental Figure S3). Immunofluorescent images in Figure 3D–I,M–O indicate that immunoreactivity for E-cadherin does not match exactly with that for N- or P-cadherin. The significance of the dominant expression of 3 kinds of cadherins in some CPE of autopsied human brains is unknown, whereas the expression of 3 kinds of cadherins in 8–10 week old mice seemed to be evenly distributed in CPE. It may represent some disabling invasion in CPEs of diseased human brains. Clinicopathological profiles of autopsied human brains used in this study are shown in Table 1, whereas the ratios of immunostained epithelial cells to total epithelial cells examined are shown in Table 4. Statistical analyses indicate a significant correlation between the ratios of immunopositive cells for E-cadherin and β-catenin but not N-cadherin nor P-cadherin. The statistical results supported the existence of E-cadherin expression in the lateral membrane of CPE. The effects of aging, neurodegeneration, and vascular accidents on the ratios were not clear. Low values of the ratios of immunostained cells in patient No. 4 may be a result of epithelial injury assumed to be caused by intraventricular hemorrhage due to thalamic hemorrhage.

In addition, immunoreactivity for E-cadherin coexisted with immunoreactivities for β-catenin in the lateral membrane of human CPE, whereas immunoreactivities for E-cadherin and vimentin were sometimes unevenly distributed on the lateral membrane in other CPEs (Figures 2A,B,E,F and 4A–F). Immunohistochemical examination using ten autopsied human brains in this study revealed that immunoreactivities for cadherins were sometimes weak or had disappeared in some epithelial cells just above calcified or densely fibrous materials. Extensive fibrosis with calcification occurred in the choroid plexus stroma

of aged brains [23], suggesting a disturbed flow of interstitial fluids including ions and nutrients in the choroid plexus stroma. CPE plays significant roles in the secretion of cerebrospinal fluid [1,2], transport of nutrients such as glucose [10,22,28], and regulation of behavioral circadian rhythms [4]. Accordingly, it is likely that various brain disorders due to abnormalities of the choroid plexus may occur in elderly brains. Circadian rhythm dysfunction which was reported in patients with senile dementia of the Alzheimer's type [5], may at least partially be due to choroid plexus dysfunction.

Cadherins are a group of genetically related glycoproteins involved in $Ca^{2+}$-dependent cell-cell adhesion [32–34]. Classical cadherins are subdivided into several classes: epithelial (E-), neuronal (N-), placental (P-), and retinal (R-) cadherins. These were the first members of the superfamily to be identified as CDH1, CDH-2, CDH-3, and CDH-4, respectively [35]. They are calcium-binding proteins with a cytoplasmic domain and binding sites for p120-catenin and β-catenin. Also, in this study, β-catenin was localized to interepithelial areas with E-cadherin. N-cadherin is well-known to be the most commonly expressed cadherin in the central nervous system and plays a key role in guiding the morphogenesis of neural tissues [16,34]. E-cadherin is expressed in all mammalian epithelia, being co-expressed and located at the cell membrane and organizing adherens junctions [35]. P-cadherin was reported as a cadherin molecule most abundant in the developing mouse placenta [32,33]. P-cadherin expression is also known to be restricted to the basal layers of stratified epithelial tissues [36,37]. Concerning their relationship, N- and E-cadherins share structural and functional characteristics and show a mutually exclusive expression pattern during embryonic morphogenesis [34]. P-cadherin is co-expressed with E-cadherin in embryonic stem cells and several adult epithelial tissues, including the breast, prostate, several organs of the digestive and urinary tracts, lung, and endometrium [38,39]. It has been suggested that P-cadherin partially overlaps with E-cadherin expression, possibly reflecting partial redundancy [35]. Also, in this study, immunohistochemical expression of E- and N-cadherins was unevenly distributed in some epithelial cells, whereas immunohistochemical expression of E-cadherin frequently overlapped with P-cadherin expression. Accordingly, these findings in the human CPE were compatible with those in other organs reported previously [35,38,39]. Interestingly, immunoreactivity for P-cadherin, indicated by a dotted arrow in Figure 3G, was not seen in the same membrane in Figure 3H. Accordingly, E-cadherin expression did not completely overlap with P-cadherin expression, as also shown in Table 4. P-cadherin is known to be poorly expressed in normal intestinal epithelial cells but upregulated in inflamed and injured mucosa [40]. Alteration of the expression of cadherins with aberrant P-cadherin expression is also known to be associated with co-expression or loss of E-cadherin expression in the gastrointestinal tract with neoplastic development or being repaired [41]. Accordingly, uneven distribution of these cadherins may be related to the E-cadherin to P-cadherin switch, happening in inflamed or injured tissues [40–42]. A large-scale study including inflamed or injured choroid plexus tissues is required to prove the hypothesis. On the other hand, E-cadherin immunoreactivity overlap with immunoreactivity for β-catenin, whereas it was unevenly distributed with immunoreactivity for vimentin, a mesenchymal cytoplasmic protein. Heterogenous expression of the mesenchymal markers in CPE was interesting, as it was similar to the epithelial-to-mesenchymal transition phenomenon [35]. Additional studies are needed to clarify whether the localization of E-cadherin in the lateral membrane of CPE is related to an epithelial-to-mesenchymal transition-like phenomenon. Calcium signaling is known to be pivotal to the circadian clock in the suprachiasmatic nucleus [43]. It is an issue worthy of further study whether calcium-dependent adherens junctional proteins in CPE are related to circadian rhythm formation. At present, however, the significance, except for cell adhesion, of the expression of E-, N-, and P-cadherins in CPE is unclear. It is unique that E-cadherin, a representative epithelial marker, exists cooperatively or exclusively with N-cadherin, a non-epithelial cadherin, in the central nervous system showing predominant expression of N-cadherin. Further studies are needed to clarify the significance of cadherins in CPE. According to the website of The Human Protein Atlas (Human Protein

Atlas proteinatlas.org, accesses on 17 August 2023), CDH1 is introduced as expressed in the human choroid plexus [44].

## 5. Conclusions

RT-PCR analysis revealed that mRNA of E-cadherin was expressed in the choroid plexus tissues of mice. Immunoreactivity for E-cadherin was localized to the lateral membrane of CPE in mice and humans. Western blotting confirmed the specificity of immunoreactivity for E-cadherin in choroid plexus tissues of mice. Immunohistochemical expression of E- and N-cadherins or vimentin was unevenly distributed in some epithelial cells, whereas immunoreactivity for E-cadherin was frequently consistent with that for P-cadherin or β-catenin in CPE of humans. Accordingly, these findings indicate that E-cadherin is expressed in the lateral membrane of CPE, possibly correlated with the expression of other cadherins and cytoplasmic proteins.

**Supplementary Materials:** The following supporting information can be downloaded at: https://www.mdpi.com/article/10.3390/cimb45100492/s1, Figure S1: Representative microphotographs of immunohistochemical staining using antibodies against E-cadherin (Protein-Tech) (A), N-cadherin (B), and P-cadherin (Santa Cruz) (C) in choroid plexus of mice. Insets of (A–C) show enlarged images of their microphotographs. Scale bars indicate 100 μm. Figure S2: The results of direct sequencing of PCR products amplified with 3 sets of Cdh1 primers (mCdh1-1 (S2a), mCdh1-2 (S2c), mCdh1-3 (S2e)) from mouse choroid plexus and their comparison with the reference (expected) sequence (S2b,d,f). The positions of primers are indicated by open arrows (red: forward, green: reverse) in S2b,d,f. The upper line "mouse Cdh1 sense" or "mouse Cdh1 antisense" indicates the reference sequence and the lower lines "mCdh1-1F," "mCdh1-1R," etc., indicate results of sequencing of PCR products presented in S2a,c,e. The position of the amplicon obtained from each primer set is indicated in the center of S2b,d,f. Figure S2b,d,f was prepared using SnapGene software (SnapGene Viewer, version 6.2, www.snapgene.com, accessed on 15 April, 2023) and CLC Sequence Viewer (version 8.0, QIAGEN Aarhus A/S). Figure S3: Microphotographs of H&E staining and immunostaining of E-cad (Dako), E-cad (ProteinTech), N-cad, and P-cad (SantaCruz) in the choroid plexus of 10 human brains (A-J). A-J are staining images of the choroid plexus in No. 1–10 shown in Table 1, respectively. Branch numbers 1, 2, 3, 4, and 5 indicate H&E staining, and immunostaining of E-cad (Dako), E-cad (ProteinTech), N-cad, and P-cad (SantaCruz), respectively. Scale bars: 100 μm.

**Author Contributions:** G.T., K.W. and M.U. performed immunostaining and confocal microscopic observation, and took pictures with microscopes; G.T. and Y.C. performed molecular analysis; Y.C., R.M., Y.M., K.M. and M.U. conducted pathological dissection; K.W., R.M., K.Y. and N.U. created figures and tables; G.T. and M.U. wrote the manuscript; Y.C., K.W., R.M., Y.M., K.M., N.U., K.Y., G.S. and Y.O. revised the manuscript critically for content. All authors have read and agreed to the published version of the manuscript.

**Funding:** Authors Y.C., R.M. and N.U. were supported by grants from JSPS KAKENHI 19K07508, 23K10827 (Y.C.), 20K16193 (R.M.), and 20K16550 (N.U.) of Japan, respectively.

**Institutional Review Board Statement:** The study using autopsied human brains conformed to the Declaration of Helsinki, and was approved by the Institutional Ethics Committee of the Faculty of Medicine, Kagawa University (#H24-48). All animal studies were approved by the Kagawa University Animal Care and Use Committee (#21636).

**Informed Consent Statement:** Informed consent was obtained from families of all human subjects for autopsy.

**Data Availability Statement:** Not applicable.

**Acknowledgments:** All authors gave approval for the final version to be published and agreed to be accountable for all aspects of the work. The authors thank T. Uyama and M. Kawauchi for database search for amino acid sequence and technical assistance, respectively.

**Conflicts of Interest:** The authors declare no conflict of interest.

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
