# Peer review of "E-Cadherin Is Expressed in Epithelial Cells of the Choroid Plexus in Human and Mouse Brains"

_cimb, doi:10.3390/cimb45100492_

Round 1

Reviewer 1 Report

This manuscript assesses the presence of E, N and P- cadherins on the epithelial cells lining the Choroid Plexus in Human and Mouse brains. The paper advances the evidence found for cadherin localization in such cells, by IHC and IF microscopy. However, with the premise that certain specific antibodies used by other studies might have cross-reacted to create a confusion in the field, these studies must be in greater detail and demonstrate real specificity of the antibodies. There can be a lot more quantification of the studies done here....to increase the usefulness and interest of the audience for CIMB and the field.

Major issues:

1. Atleast for E-cadherin, are there any KO or KD cell lines that they could test their mouse and rabbit antibodies on? The staining for E-cadherin looks convincing, both IHC and IF. But if the P-Cadherin or N-cadherin antibodies do cross react and detect some E-cadherin, or vice-versa, then the inferences drawn from the Figures might turn out not to be entirely correct. The authors could also refer to antibody datasheets that might claim specificity, but performing these experiments on their own would be stronger for the manuscript.

2. The RT-PCR derived gel in Fig 1C does not look convincing. If a RT-qPCR was run, then please provide the relative gene expression data, and not just a gel picture. Also, I am not sure why different PCR amplification conditions were used for GAPDH and Cdh1: the controls should be run in the same conditions as the sample. Providing real-time differences in gene expression can support the protein expression and immunostaining data, which should ideally be done for not only E-cadherin, but also the other cadherins studied in this paper.

3. Wherever the staining of particular cadherins look weaker in one section of cells and stronger in another, the authors refer to that as "uneven" distribution. While the plasma membrane distribution certainly looks uneven, the fact that some cells stain much brighter for one cadherin and less so for another within the group of cells shown, is interesting. This should also be highlighted, and an overall quantification as well as specific quantification of the staining should be done to elucidate the differences. Since there were adequate human tissue samples, the distribution and quantification of E-, P- and N- cadherin could then be analyzed from the IHC or the IF data if not both.

4. (Figure 4) Vimentin also looks disproportionately present in some cells over the others, and not correlating with E-cadherin. Quantifications of these could be useful in such a scenario.

no comments

Author Response

September 18, 2023

Assistant Editor

Current Issues in Molecular Biology

Manuscript ID: cimb-2608381

Dear Dr. Maxine Mei,

I wish to submit a revised version of an original article for publication in Current Issues in Molecular Biology, entitled “E-cadherin expression in epithelial cells of the choroid plexus in human and mouse brains”. The paper was coauthored by Genta Takebayshi, Yoichi Chiba, Keiji Wakamatsu, Ryuta Murakami, Yumi Miyai, Koichi Matsumoto, Naoya Uemura, Ken Yanase, Gotaro Shirakami, Yuichi Ogino, and Masaki Ueno.

I have taken all the reviewers’ comments into account and have revised the manuscript by red letters. I have read and understood your journal’s policies. There are no conflicts of interest to declare. Please consider this paper for publication in “Current Issues in Molecular Biology”. Thank you for your consideration.

Best regards,

Professor Masaki Ueno, M.D., Ph.D.

Department of Pathology and Host Defense

Faculty of Medicine, Kagawa University,

1750-1 Ikenobe, Miki-cho, Kita-gun,

Kagawa 761-0793, Japan

Fax: +81-87-891-2116, Tel.: +81-87-891-2115,

E-mail: ueno.masaki@kagawa-u.ac.jp

To reviewer 1:

Comment 1: Atleast for E-cadherin, are there any KO or KD cell lines that they could test their mouse and rabbit antibodies on? The staining for E-cadherin looks convincing, both IHC and IF. But if the P-Cadherin or N-cadherin antibodies do cross react and detect some E-cadherin, or vice-versa, then the inferences drawn from the Figures might turn out not to be entirely correct. The authors could also refer to antibody datasheets that might claim specificity, but performing these experiments on their own would be stronger for the manuscript. To (comment 1):

According to the reviewer’s comment, we present data on the datasheet for E-cadherin polyclonal antibody (ProteinTech, 20874-1-AP) that claim specificity (https://www.ptglab.co.jp/Products/E-cadherin-Antibody-20874-1-AP.htm). Western blot results of this E-cadherin antibody with sh-E-cadherin transfected A431 cells indicates a decrease in a band around 120 kDa (KD validated). Various lysates from A431, HCT116, MCF-7, and T-47D cells subjected to SDS PAGE followed by Western blot with the E-cadherin antibody show a single band around 120 kDa in each lane. This antibody has been widely used. Regarding the wide applicability of this antibody (20874-1-AP), there were 1352 publications for Western blot and 252 publications for immunohistochemistry. In addition, according to the alignment of aminoacid sequences of E-cadherin and P-cadherin from human and mouse, their identities were 54% in human and 52% in mouse. Immunogen of this antibody is E-cadherin fusion protein Ag14973, PIFNPTTYKGQVPENEANVVITTLKVTDADAPNTPAWEAVYTILNDDGGQFVVTTNPVNNDGILKTAKGLDFEAKQQYILHVAVTNVVPFEVSLTTSTATVTVDVLDVNEAPIFVPPEKRVEVSEDFGVGQEITSYTAQEPDTFMEQKITYRIWRDTANWLEINPDTGAISTRAELDREDFEHVKNSTYTALIIATDNGSPVATGTGTLLLILSDVNDNAPIPEPRTIFFCERNPKPQVINIIDADLPPI (373-622). The alignment of partial amino acid sequences of E-cadherin used as immunogen was 54% identical to that of P-cadherin.

In addition, we present data on E-cadherin monoclonal antibody (Dako, M3612, NCH-38). According to the datasheet for this monoclonal antibody (Dako)

(https://www.citeab.com/antibodies/2414873-m3612-e-cadherin-concentrate) (CiteAb) or (https://www.labome.com/product/Dako/M3612.html) (labome), this antibody has been also widely used. Regarding the wide applicability of this antibody, there were 195 (by CiteAb) or 54 (by labome) publications. The data sheet for monoclonal anti-E-cadherin antibody (Dako, M3612, clone NCH-38) certifies that the anti-E-cadherin antibody recognizes the 120 kDa mature form and 82 kDa fragment of E-cadherin in Western blots of A431 cells lysates. Immunogen of this antibody is E-cadherin (uvomorulin) and GST recombinant protein.

As we can see a single band around 120 kDa in Fig.1b, it strongly indicates that this E-cadherin monoclonal antibody (Dako, M3612) recognizes E-cadherin specifically. In addition, Table 4 does not indicate that the anti-E-cadherin antibodies recognize N- or P-cadherin. However, as the possible cross reactivity by other cadherin antibodies cannot be completely excluded judging from the findings in this study alone, we added descriptions on the specificity of E-cadherin antibodies and the possible cross reactivity of cadherin antibodies (lines 367-381).

 Comment 2: The RT-PCR derived gel in Fig 1C does not look convincing. If a RT-qPCR was run, then please provide the relative gene expression data, and not just a gel picture. Also, I am not sure why different PCR amplification conditions were used for GAPDH and Cdh1: the controls should be run in the same conditions as the sample. Providing real-time differences in gene expression can support the protein expression and immunostaining data, which should ideally be done for not only E-cadherin, but also the other cadherins studied in this paper. To (comment 2):In this study, we performed reverse-transcription PCR (RT-PCR), but not real-time RT-PCR, to just show the presence or absence of mRNA of E-cadherin, since mRNA and protein expressions of N-cadherin and P-cadherin had been confirmed in epithelial cells of choroid plexus in other studies. To enable rough comparison of relative mRNA expression level by comparing band intensity, we presented gel images of amplicons of Cdh1 from 3 pairs of specific primers set at different positions of Cdh1 cDNA, and those of Gapdh as an internal standard in Fig. 1C. Due to the difference in the expression level between Cdh1 and Gapdh, we presented amplicons with different cycle numbers: Gapdh ampicon signals are saturated at 33 cycles of PCR reaction, whereas Cdh1 amplicon signals are not visible at 24 cycles of reaction. Comment 3: Wherever the staining of particular cadherins look weaker in one section of cells and stronger in another, the authors refer to that as "uneven" distribution. While the plasma membrane distribution certainly looks uneven, the fact that some cells stain much brighter for one cadherin and less so for another within the group of cells shown, is interesting. This should also be highlighted, and an overall quantification as well as specific quantification of the staining should be done to elucidate the differences. Since there were adequate human tissue samples, the distribution and quantification of E-, P- and N- cadherin could then be analyzed from the IHC or the IF data if not both. To (comment 3):

According to the reviewer’s comment, we performed morphometrical analyses of immunostained sections of human brain samples. The percentage of epithelial cells positive for immunostaining of E-cadherin (Dako), E-cadherin (Proteintech), N-cadherin, P-cadherin (SantaCruz), b-catenin, and vimentin in ten randomly selected areas in the choroid plexus were examined using ten autopsied human brains. Mean values of the percentage were calculated. Accordingly, we added descriptions on the morphometry in Material and Methods (lines 216-222), Results (lines 331-348), and Discussion (lines 389-397). In addition, we newly made Table 4, in which mean values of the percentage of the immunostained cells was calculated. The results of morphometrical analyses supported the existence of E-cadherin in the lateral membrane of CPE.

Comment 4:

(Figure 4) Vimentin also looks disproportionately present in some cells over the others, and not correlating with E-cadherin. Quantifications of these could be useful in such a scenario.

To (comment 4):

According to the reviewer’s comment, we performed morphometry also using an antibody for vimentin and reported findings on mean values of the percentage of epithelial cells immunostained with the antibody for vimentin in Table 4. The statistical analyses showed significant correlation in expression between vimentin and E-cadherin (Dako) (p = 0.02) but not E-cadherin (ProteinTech) (p = 0.15). Description on morphometrical analyses on vimentin expression was added (lines 331-348).

Reviewer 2 Report

The goal of this paper is to investigate expression and localization of major classical cadherins in epithelial cells of the human and murine Choroid Plexus. The authors used a combination of immunofluorescence labeling, immunoblotting and RT-PCR and focused on three major cadherins, CDH1 (E-cad), CDH2 (N-cad) and CDH3 (P-cad). Overall, the study has scientific merit, and the manuscript sows some interesting results. There are a few minor issues that need to be addressed prior to its publication.

Comments: 

1. The CDH1 amplicons signals are barely visible in Figure 1C. I wonder if the authors could provide images with higher exposure.

2. The specificity and possible cross reactivity of cadherin antibodies used in this study should be discussed. This was an issue for the previous studies that tried to localize cadherins in the brain.

3. High expression of P-cadherin in choroid plexus is interesting, but I wonder if some of the data reflect the fact that tissue samples of patients with different types of inflammation have been used. There is an interesting phenomenon that is called 'E-cadherin to P-cadherin switch, happening in inflamed tissues. The authors should discuss this phenomenon and cite appropriate literature such as: Sanders DS et al, J. Pathol. 2000, 190: 526-530; Naydenov NG et al, Cells 2022, 11:1467; Zbar AP J Gastroenterol. 2004, 39:413-421.

4. A few types in the introduction. For example, line 53, 'occludin' should be singular. Line 72, P-cadherin?

Author Response

September 18, 2023

Assistant Editor

Current Issues in Molecular Biology

Manuscript ID: cimb-2608381

Dear Dr. Maxine Mei,

I wish to submit a revised version of an original article for publication in Current Issues in Molecular Biology, entitled “E-cadherin expression in epithelial cells of the choroid plexus in human and mouse brains”. The paper was coauthored by Genta Takebayshi, Yoichi Chiba, Keiji Wakamatsu, Ryuta Murakami, Yumi Miyai, Koichi Matsumoto, Naoya Uemura, Ken Yanase, Gotaro Shirakami, Yuichi Ogino, and Masaki Ueno.

I have taken all the reviewers’ comments into account and have revised the manuscript by red letters. I have read and understood your journal’s policies. There are no conflicts of interest to declare. Please consider this paper for publication in “Current Issues in Molecular Biology”. Thank you for your consideration.

Best regards,

Professor Masaki Ueno, M.D., Ph.D.

Department of Pathology and Host Defense

Faculty of Medicine, Kagawa University,

1750-1 Ikenobe, Miki-cho, Kita-gun,

Kagawa 761-0793, Japan

Fax: +81-87-891-2116, Tel.: +81-87-891-2115,

E-mail: ueno.masaki@kagawa-u.ac.jp

To reviewer 2:

Comment 1:

The CDH1 amplicons signals are barely visible in Figure 1C. I wonder if the authors could provide images with higher exposure.

To (comment 1):

Thank you for your comment. According to the reviewer’s comment, we submitted a clearer new image of the CDH1 amplicons signals in Fig. 1C in the revised version.

Comment 2:

The specificity and possible cross reactivity of cadherin antibodies used in this study should be discussed. This was an issue for the previous studies that tried to localize cadherins in the brain.

To (comment 2):

We present data on two kinds of antibodies for E-cadherin before we discuss the specificity and possible cross reactivity of E-cadherin antibodies.

First, we present data on the datasheet for E-cadherin polyclonal antibody (ProteinTech, 20874-1-AP) that claim specificity (https://www.ptglab.co.jp/Products/E-cadherin-Antibody-20874-1-AP.htm). Western blot results of this E-cadherin antibody with sh-E-cadherin transfected A431 cells indicates a decrease in a band around 120 kDa (KD validated). Various lysates from A431, HCT116, MCF-7, and T-47D cells subjected to SDS PAGE followed by Western blot with the E-cadherin antibody show a single blot around 120 kDa in each lane. This antibody has been widely used. Regarding the wide applicability of this antibody (20874-1-AP), there were 1352 publications for Western blot and 252 publications for immunohistochemistry. In addition, according to the alignment of aminoacid sequences of E-cadherin and P-cadherin from human and mouse, their identities were 54% in human and 52% in mouse. Immunogen of this antibody is E-cadherin fusion protein Ag14973 and the alignment of partial amino acid sequences of E-cadherin used as immunogen was 54% identical to that of P-cadherin.

Next, we present data on E-cadherin monoclonal antibody (Dako, M3612, NCH-38). According to the datasheet of this monoclonal antibody (Dako)

(https://www.citeab.com/antibodies/2414873-m3612-e-cadherin-concentrate) (CiteAb) or (https://www.labome.com/product/Dako/M3612.html) (labome), this antibody has been also widely used. Regarding the wide applicability of this antibody (Dako), there were 195 (by CiteAb) or 54 (by labome) publications. The data sheet for monoclonal anti-E-cadherin antibody (Dako, M3612, clone NCH-38) certifies that the anti-E-cadherin antibody (Dako, M3612, clone NCH-38) recognizes the 120 kDa mature form and 82 kDa fragment of E-cadherin in Western blots of A431 cells lysates. Immunogen of this antibody (Dako, M3612) is E-cadherin (uvomorulin) and GST recombinant protein.

As we can see a single band around 120 kDa in Fig.1b, it strongly indicate that this E-cadherin monoclonal antibody (Dako, M3612) recognizes E-cadherin specifically. Table 4 does not indicate that the anti-E-cadherin antibodies recognize N- or P-cadherin and rather supports the existence of E-cadherin in the choroid plexus. However, the possible cross reactivity of cadherin antibodies cannot be completely excluded judging from experimental data in this study alone. Accordingly, we added descriptions on the specificity of E-cadherin antibodies and described that the possible cross reactivity cannot be completely excluded judging from the findings in this study alone (lines 367-381).

Comment 3:

High expression of P-cadherin in choroid plexus is interesting, but I wonder if some of the data reflect the fact that tissue samples of patients with different types of inflammation have been used. There is an interesting phenomenon that is called 'E-cadherin to P-cadherin switch, happening in inflamed tissues. The authors should discuss this phenomenon and cite appropriate literature such as: Sanders DS et al, J. Pathol. 2000, 190: 526-530; Naydenov NG et al, Cells 2022, 11:1467; Zbar AP J Gastroenterol. 2004, 39:413-421.

To (comment 3):

Thank you so much for your kind suggestions. We discussed the ‘E-cadherin to P-cadherin switch’ in the Discussion (lines 437-444) and cited new three references as [40, 41, 42] (lines 599-605).

Comment 4:

A few types in the introduction. For example, line 53, 'occludin' should be singular. Line 72, P-cadherin?

To (comment 4):

According to the reviewer’s comment, “occludins” (line 53) was changed to singular “occludin”.

On the other hand, molecular cloning and characterization of B-cadherin was reported as a novel chick cadherin (J Cell Biol 1991, 113, 893-905). Although B-cadherin expression was not reported in human CPE, the cytoplasmic domains of B-cadherin and E-cadherin were reported to be 88% identical in chickens [20]. Accordingly, the similarities between them were described. However, as this sentence on similarities in expression between E-cadherin and B-cadherin was difficult to understand in the original version, it was modified in the revised version (lines 73-75).
